# Workers’ Perception Heat Stress: Results from a Pilot Study Conducted in Italy during the COVID-19 Pandemic in 2020

**DOI:** 10.3390/ijerph19138196

**Published:** 2022-07-04

**Authors:** Michela Bonafede, Miriam Levi, Emma Pietrafesa, Alessandra Binazzi, Alessandro Marinaccio, Marco Morabito, Iole Pinto, Francesca de’ Donato, Valentina Grasso, Tiziano Costantini, Alessandro Messeri

**Affiliations:** 1Occupational and Environmental Medicine, Epidemiology and Hygiene Department, Italian Workers’ Compensation Authority (INAIL), 00143 Rome, Italy; m.bonafede@inail.it (M.B.); e.pietrafesa@inail.it (E.P.); a.binazzi@inail.it (A.B.); a.marinaccio@inail.it (A.M.); 2Epidemiology Unit, Department of Prevention, Local Health Authority Tuscany Centre, 50135 Florence, Italy; miriam.levi@uslcentro.toscana.it; 3Institute of Bioeconomy, National Research Council (IBE-CNR), 50019 Florence, Italy; marco.morabito@ibe.cnr.it; 4Physical Agents Sector, Regional Public Health Laboratory, 53100 Siena, Italy; iole.pinto@uslsudest.toscana.it; 5Department of Epidemiology Lazio Regional Health Service, ASL ROMA 1, 00147 Rome, Italy; f.dedonato@deplazio.it (F.d.D.); t.costantini@deplazio.it (T.C.); 6LaMMA Consortium—Weather Forecaster and Researcher at Laboratory of Monitoring and Environmental Modelling for Sustainable Development, 50019 Florence, Italy; valentina.grasso@ibe.cnr.it; 7Climate and Sustainability Foundation, 50100 Florence, Italy; 8AMPRO—Professional Weather Association, 00142 Rome, Italy

**Keywords:** risk perception, risk knowledge, heat stress prevention measures, heat exposure, occupational injuries

## Abstract

Many workers are exposed to the effects of heat and often to extreme temperatures. Heat stress has been further aggravated during the COVID-19 pandemic by the use of personal protective equipment to prevent SARS-CoV-2 infection. However, workers’ risk perception of heat stress is often low, with negative effects on their health and productivity. The study aims to identify workers’ needs and gaps in knowledge, suggesting the adaptation of measures that best comply with the needs of both workers and employers. A cross-sectional online questionnaire survey was conducted in Italy in the hottest months of 2020 (June–October) through different multimedia channels. The data collected were analyzed using descriptive statistics; analytical tests and analysis of variance were used to evaluate differences between groups of workers. In total, 345 questionnaires were collected and analyzed. The whole sample of respondents declared that heat is an important contributor to productivity loss and 83% of workers did not receive heat warnings from their employer. In this context, the internet is considered as the main source of information about heat-related illness in the workplace. Results highlight the need to increase workers’ perception of heat stress in the workplace to safeguard their health and productivity. About two-thirds of the sample stated that working in the sun without access to shaded areas, working indoors without adequate ventilation, and nearby fire, steam, and hot surfaces, represent the main injuries’ risk factors.

## 1. Introduction

Mean annual air temperatures are getting hotter globally due to climate change [1]. The year 2021 was the 7th consecutive year (2015–2021) where the global temperature had been over 1 °C above pre-industrial levels (1850–1900), with 2016, 2019, and 2020 constituting the top three ones [2,3]. Because of climate change, a substantial increase in the frequency and intensity of heat waves has been observed in the hottest months of the year, and it has been estimated that around 30% of the world population is currently exposed to climatic conditions particularly critical for human health for at least 20 days a year [4]. Workers, in particular those who spend most of their activities outdoors, are among the individuals the most exposed to the effects of heat and in general to extreme temperatures [5,6]. The situation has further deteriorated during the current COVID-19 pandemic due to the widespread use of personal protective equipment (PPE) to prevent SARS-CoV-2 infection, which tends to increase heat stress [7,8,9]. The challenges derived from heat exposure to workers’ health and productivity [10] have already been identified as significant problems in tropical areas and are becoming more and more common also in the USA and in EU countries; not only outdoor workers, such as farmers and construction workers [11,12], but also indoor workers performing tasks nearby heat-generating equipment [13,14], such as iron and steel workers, boiler room workers, bakers, firefighters, especially if involved in moderate or high-intensity activities, are at the higher risk of heat illnesses, injuries, and even heat stress-related death [15].

Occupational heat stress is a risk factor for medical conditions collectively defined as heat illnesses, which include minor symptoms such as heat rash, heat cramps, and heat edema, and more serious conditions such as heat syncope and heat exhaustion [4]. The most severe form of heat illness is heatstroke. Contrary to a classic heatstroke, which more commonly occurs among the elderly, children and people with underlying chronic diseases, the exertional heatstroke, the one occurring among workers, typically affects healthy young individuals. Heatstroke is a potentially life-threatening health condition that is facilitated by carrying out strenuous activities in severe heat and/or humidity [16]. Kidney diseases are also often diagnosed in otherwise healthy young adults commonly exposed to heat and dehydration in the workplace [17,18].

Heat-related illnesses and injuries are largely preventable. It is essential that workers know the possible health effects of working in the heat and that heat-illness prevention and response programs are established in the workplace so that workers are kept safe from the health effects of extreme heat.

There is a need to investigate the baseline information regarding how people perceive the heat risk to develop a heat stress effective management system. Workers’ awareness of the possible effects of heat stress and perceptions of its risk also constitute an essential part of policy decisions and improving climate change risk information and communication [19,20,21].

In Italy, the WORKLIMATE project (“Impact of environmental thermal stress on workers’ health and productivity: intervention strategies and development of an integrated heat and epidemiological warning system for various occupational sectors”, https://www.worklimate.it) (accessed on 30 June 2022), which started in June of 2020, has the aim to improve the knowledge base and awareness among workers on the health effects of environmental thermal stress conditions. As part of the project activities, a web-based questionnaire survey was conducted at the national level to investigate workers’ perceptions and knowledge regarding the negative consequences of occupational heat stress, especially during COVID-19, and to identify potential barriers to prevent heat-related illnesses in the workplace, including education and training. The ultimate goal of our study is to identify workers’ needs and gaps in knowledge, suggesting the adaptation measures that best comply to the needs of both workers and employers.

## 2. Materials and Methods

A cross-sectional questionnaire survey was conducted in Italy among workers in the hottest months of 2020, from the 1st of June to the 31st of October, through different multimedia channels, in order to reach a wide and varied target at the national level, specifically the following platforms were used: Physical Agents Portal (https://www.portaleagentifisici.it/) (accessed on 30 June 2022), Facebook, Twitter, LinkedIn, and WhatsApp, based on a communication plan daily updated. Direct mailing was used as well. The questionnaire was distributed through the Google Form online platform (https://docs.google.com/forms/d/19R5EGY5nH6k5vsjEAtx5Hx__SiV1l4Iv5BieHsV2m1U/edit?ts=5f0c33c5, last accessed on 11 January 2022), complemented by an informed consent form. Participation was voluntary and anonymous. The estimated completion time was around 20 min. Data were collected, stored, and analyzed according to the Regulation on the protection of natural persons with regard to the processing of personal data (EU Regulation 2016/679-General Data Protection Regulation-GDPR-application from 25 May 2018). This activity received the ethical clearance from the Commission for Ethics and Integrity of Research of the National Research Council (CNR) (N. 0009389/2020, 2 June 2020).

### 2.1. Questionnaire Design

The questionnaire of this pilot study (Appendix A) was constructed ad hoc, taking into consideration the main literature review on the subject [22,23,24,25,26,27,28,29,30,31]. A pre-testing on a random sample allowed the optimization of the instrument and to determine the time needed to complete the questionnaire.

The survey is composed by four sections:SECTION A—DEMOGRAPHIC AND SOCIO-OCCUPATIONAL DATA—gender, age, school degree qualification, nationality, fasting for personal reasons, geographical area of work, work environment, marital status, number of children, job sector, job performed, company size, physical activity, presence of heat sources, use of chemicals, use of protective clothing, use of COVID-19 masks, warm months of the year worked, experience in Occupational Safety and Health (OSH), diagnosis of infection with the SARS-CoV-2 virus, development of COVID-19 disease in symptomatic form, and the presence of chronic diseases (questions from 1 to 25);SECTION B—RISK PERCEPTION—questions on the qualitative dimensions of the risk [29,30,31] associated with heat stress, i.e., general risk perceived, voluntary nature, immediacy of effects, personal knowledge, scientific knowledge, novelty, chronic/catastrophic, common/terrifying, future generations, control of severity, visibility, personal exposure, collective exposure, severity of consequences (questions 26 to 43 on a 5-point Likert scale from 1 = “strongly disagree” to 5 = “strongly agree”);SECTION C—RISK KNOWLEDGE—questions on the evidence relating to the most important effects of heat waves and heat stress, the categories of workers involved, and the main factors of vulnerability (questions 44 to 57 on a 5-point Likert scale from 1 = “strongly disagree” to 5 = “strongly agree”);SECTION D—ACCIDENTS, PREVENTION MEASURES AND WORK POLICIES—questions about the frequency of heat-related diseases and injuries, opinions about work factors/hazards, and organizational aspects that contribute to the occurrence of such injuries, types of workers involved, heat injury prevention training, main sources of information on the prevention of heat-related diseases and injuries, warnings or alerts about the possibility of a heat wave, perception of loss of productivity, perceived obstacles to prevent heat-related workplace injuries (questions 58 to 81).

### 2.2. Study AREA and Climatic Characteristicsg

In the period of the questionnaire administration (from June to October 2020), during the complex management of the COVID-19 pandemic, climatic conditions in Italy were characterized by air temperatures generally above the average compared to the reference period 1981–2010. In particular, the most important thermal anomalies occurred in central Italy (Figure 1A), with positive anomalies close to 1.5 °C compared to 1981–2010. Concerning to the two hottest summer months (July and August), July (Figure 1B) revealed the highest thermal anomalies, greater than 1.0 °C compared to the climatological average in central and southern Italy, with peaks of 1.2 °C in Lazio and Campania regions. In August (Figure 1C), the thermal anomaly decreased, however, maintaining temperatures between 0.6 and 1.0 °C above the average compared to 1981–2010.

### 2.3. Data Analysis

The data collected were analyzed using descriptive statistics (i.e., frequency, mean, standard deviation) and analytical tests. The analysis of variance (ANOVA) and chi-square analysis (χ^2^) were used to evaluate differences between groups. The chosen groups (for example, age, school degree qualification, workplace environment, use of wearing protecting clothing, use of COVID 19 mask, chronic diseases, etc.) were further grouped into three macro-groups (a. Demographic and professional characteristics, b. Characteristics of the work, c. Factors aggravating heat stress) in order to evaluate the fundamental aspects in the assessment of risk perception. The homogeneity of variance was verified with Levene’s test. The Brown–Forsythe and Welch tests were used when the homogeneity of variance assumption did not hold for the data. A Principal Component Analysis (PCA) with Varimax rotation was carried out and Cronbach’s Alpha calculation allowed an empirical assessment of the reliability to assess the dimensionality of sections “RISK PERCEPTION” and “RISK KNOWLEDGE”. The results were considered significant at a *p*-value less than 0.05. All analyses were performed using SPSS v.25.0 for Windows (IBM, Armonk, NY, USA).

## 3. Results

### 3.1. Descriptive Analysis

In total, 345 workers participated in the self-administered web survey, most of whom (67.5%) carried out their work activities in central Italy. The sex distribution was coherent with that of the employed population in Italy with 57.7% men. The average age of participants was 45.4 years (SD ± 10.7): 59.7% of the sample in their professional life are or have been involved in OSH and 66.7% of the sample suffer from chronic diseases. The level of education (school degree qualification) of the respondents was high, with 61.2% of them having a bachelor/specialist/postgraduate degree and 30.4% of them having a high school diploma. As regards to the working environment, 64.9% of workers were mainly indoors in an air-conditioned environment, 21.2% were mainly indoors in a non-air-conditioned environment, and 13.9% of them were mainly outdoors. The most represented occupational sectors were professional, scientific, and technical activities (25.2%); construction (15.7%); public administration and Armed forces/military (11.9%); manufacturing (8.1%); and health and social works (8.1%). One in four (25.5%) received training on the prevention of heat-related injuries in the workplace, and 17.1% received warnings or alerts (Table 1).

The main sources of information on the prevention of heat-related illness in the workplace were internet (16%), specific training in the workplace (13.8%), occupational physician (11.2%), TV and radio (8.4%) (Figure 2).

The whole sample perceived that heat is an important contributor to productivity loss (m = 3.93 on a scale of 1 to 5) (Figure 3).

In total, 64.6% of the respondents stated that rarely or sometimes or often injuries occur (at least partly) due to hot/high humidity conditions (Table 1). According to this group of workers, the factors/risks that contributed most to the occurrence of these heat-related injuries/illnesses were working in the sun without access to shade (solar radiation) (m = 3.97, sd = 1.04 on Likert scale from 1 = not at all to 5 = fully); working indoors without air conditioner, fan, or adequate ventilation (m = 3.74, sd = 1.08); and fire, steam, hot surfaces (m = 3.69, sd = 1.15). Again, for the same respondents, the organizational aspect mostly contributing to the occurrence of these heat-related injuries/illnesses was the lack of specific health and safety training on heat stress (m = 3.58, sd = 1.17 on Likert scale from 1 = not at all to 5 = fully). The workers who had mostly suffered these heat injuries were those between the ages of 56 and 65 (30.1%) and those over 65 (24.9%).

### 3.2. Principal Component Analysis of Section Risk Perception

A Principal Components analysis (PCA) was carried out on “Risk perception” to verify the existence of common dimensions. Four factors that explain 64.1% of the variance emerged from the analysis (Table 2).

The first factor (α = 0.83), which explains the 30.3% of the variance, was called “Personal exposure and fear of risk”, because it brings together all the items concerning personal exposure to heat risk and related fear.

The second factor (α = 0.69), which explains the 14.3% of the variance, was called “Collective exposure and risk quality”, because it brings together all the items concerning collective exposure to hot risk and the general qualities of this risk such as immediate effect, chronic or catastrophic nature, and voluntariness.

The third factor (α = 0.52), which explains the 10.5% of the variance, was called “Impact on health and prevention”, because it brings together all the items concerning how much prevention measures in the workplace can reduce risk severity and the existence of observable symptoms.

The fourth factor (α = 0.40), which explains the 9.0% of the variance, was called “Knowledge risk perception”, because it brings together all the items concerning opinions on the degree of knowledge of heat risk by workers and the scientific world.

In the factorial solution, the items 26, 27, 32, 34, 35, 43 were excluded.

### 3.3. Principal Component Analysis of Section Risk Knowledge

Principal Component Analysis (PCA) was carried out on items of “Risk knowledge” to verify the existence of common dimensions. One factor (α = 0.83), which explains the 54.4% of the variance, emerged from the analysis (Table 3).

In the factorial solution the items 46, 47, 51, 52, 53, 55, 56, 57 were excluded.

### 3.4. Risk Perception: Differences between Groups

Table 4 shows the results reported by the respondents for the section “Risk perception”.

Regarding the factor “Personal exposure and fear of risk”, and in particular, the macro groups “Demographic and professional characteristics” (a), “Characteristics of the work” (b), and “Factors aggravating heat stress” (c) (Table 5), the respondents considered themselves to be exposed to heat on average (item 38).

The feeling of being particularly exposed to heat risk was associated with: a lower level of education (school degree qualification); working outdoors or indoors in a non-air-conditioned environment; a high or very high work intensity; working near heat sources or use chemicals; wearing protective clothing; wearing a COVID mask for more than 5 h. During a heat wave, the sample felt on average at risk (item 39), in particular, those with a lower education, those suffering from chronic diseases, those working mainly outdoors. The entire sample had little fear of personally being the victim of an accident at work caused by heat waves (item 41). The most afraid were those who have been doing the same job for more than 20 years, those who work mainly outdoors, those who have a high or very high work intensity, those who work near heat sources or use chemicals, and those who wear protective clothing. The responding workers also had little fear of getting sick from heat waves (item 42), more fear was felt by those who work mainly outdoors.

Regarding the factor “Collective exposure and risk quality”, respondents thought that during a heat wave in Italy, there are many workers at risk (item 40), in particular, those suffering from chronic diseases. The sample agreed on average, that heat risk is involuntary (item 28) and that it represents a potentially lethal risk (item 33). There was little agreement among the sample with the statement “Heat causes an immediate fatal effect for those exposed” (item 29).

Regarding the factor “Impact on health and prevention”, the respondents believed that preventive measures in the workplace can reduce the severity of heat risk (item 36), in particular, it was stated by those with a higher education, those who work mainly indoors in air-conditioned and non-air-conditioned environments, those with a light or very light work intensity, those who do not work near heat sources, those who do not use protective clothing. The sample considered the average observable thermal damage, i.e., that the symptoms of injuries or illnesses due to exposure to heat are on average recognizable (item 37).

Regarding the factor “Knowledge risk perception”, according to the whole sample, the scientific community has quite little knowledge about heat risk (item 31), especially younger people (up to 40 years old), those who do not work or have worked on OSH, those who do not receive heat risk warnings, those who have not received training on heat injury prevention. The entire sample agreed that workers exposed to heat have little knowledge of the risk (item 30), in particular, those who have a higher education, those who work mainly indoors in an air-conditioned environment, those who have a light or very light work intensity, those who do not receive heat risk warnings, those who have not received training on the prevention of heat-related injuries, those who do not work near heat sources.

### 3.5. Risk Knowledge: Differences between Groups

The responses related to risk knowledge were re-coded in ‘correct’ and ‘incorrect’ knowledge.

The entire sample shows little knowledge of hot-weather risk. The only questions answered correctly by more than 40% were: “Due to the shade of the buildings, heat waves are less common in cities than in rural areas” (51.9%), “Heat stress during the night is of no importance” (59.4%), “Heat waves can be a risk factor for depression and anxiety” (44.9%). As for the first statement, the opposite is true. The second question was answered more correctly by women (68.5%, *p* = 0.002), those who do not work near heat sources (62.9%, *p* = 0.004), those who have not received training on the prevention of heat injuries (62.6%, *p* = 0.025).

Questions answered less than 20% correctly were: “Heat can cause injuries for those working in an unconditioned indoor environment” (16.2%), “Younger workers are particularly vulnerable during a heat wave” (6.1%), “Excessive sweating during a heat wave can be a sign of heat stress” (19.4%), ‘Heat waves promote the growth of harmful bacteria in water and food’ (18.6%).

### 3.6. Perceived Obstacles to Preventing Heat-Related Workplace Injuries: Differences between Groups

Respondents believed that the top five obstacles to preventing heat-related occupational accidents (Figure 4) were:Lack of commitment by employers to protect health and safety (m = 3.92, sd = 1.14 on a scale of 1 to 5); particularly for those with chronic illnesses (m = 4.15, sd = 1.06, F = 7.28, *p* = 0.007) and those who have not received training on preventing heat-related injuries (m = 4.02, sd = 1.10, F = 9.17, *p* = 0.003).Lack of training by company health and safety managers (m = 3.91, sd = 1.13); especially of those who have not received training on preventing heat-related injuries (m = 4.04, sd = 1.04, F = 10.19, *p* = 0.002) and those working in large companies (m=4.12, sd = 1.06, F = 3.26, *p* = 0.022).Lack of training of workers (m = 3.81, sd = 1.12); especially of those with higher education (m = 3.96, sd = 1.04, F = 8.85, *p* = 0.003), those not trained in heat injury prevention (m = 3.94, sd = 1.06, F = 13.26, *p* = 0.000), and those working in large companies (m = 4.02, sd = 1.08, F = 3.23, *p* = 0.023).Lack of compliance with regulations (m = 3.79, sd = 1.07); especially for those working in medium-sized (m = 3.98, sd = 1.02, F = 5.12, *p* = 0.002) and large companies (m = 3.92, sd = 1.08, F = 5.12, *p* = 0.002), those suffering from chronic illnesses (m = 3.97, sd = 1.00, F = 5.44, *p* = 0.020).Lack of awareness among company health and safety managers of the risks from heat (m = 3.77, sd = 1.18); especially for women (m = 3.98, sd = 1.05, F = 8.25, *p* = 0.004) and those who have not received training on preventing heat-related injuries (m = 3.94, sd = 1.06, F = 16.79, *p* = 0.000).

## 4. Discussion

The year 2020 was the second hottest year on Earth in a record 140 years (just behind 2016) and the hottest year on record in Europe [32]. An increasing number of epidemiological studies have provided evidence of the association between heat exposure and the risk of accidents at work [5,6,14,23,33,34,35] and this phenomenon can be explained by a decrease in cognitive performance in people who work in hot and humid environments in Europe [36]. Confirming this aspect, a recent review demonstrated that a raised core temperature is associated with a reduction in vigilance and more complex dual-task performance [37]. In addition, also dehydration associated with hot conditions causes a severe reduction in physical and cognitive performance [37,38,39,40]. In general, according to Varghese et al. [35], work-related injuries/accidents in hot conditions can be caused by physical discomfort and altered behavior, fatigue, declining psycho-motor performance, loss of concentration, and reduced alertness.

Prolonged exposure to heat can also have a major impact on productivity [34,41,42,43]. A better understanding of how workers perceive the risks of exposure to heat in the workplace is necessary for the development of heat prevention strategies [35] and to minimize the impact of extremely high temperatures on the health and safety of workers [44]. However, only a few studies have investigated perceptions of heat risk among workers [9,19,21,22,24,25,26,27,45,46].

The main strength of this study is that the increase of knowledge of the heat risk workers’ perception can be particularly useful for the development of the risk awareness process by all safety actors. The results of this study showed that the categories most exposed to heat risk are those who feel most at risk, even during a heat wave, and who are most afraid of being personally the victim of an accident at work caused by heat waves or getting sick from it. This result confirms the evidence of the Australian survey [19,27,46] and more generally of the more developed countries.

The whole sample considered that during a heat wave in Italy, there are many workers at risk, and that on average heat risk is involuntary and potentially lethal. However, it emerged that the risk perception was low in younger workers (less than 40 years old), in contrast to what emerged in the recent study on the general population in Urban Citizen in Germany [24], where highest heat risk perception was among people aged 18–29 years. Our result is in line with what emerged in Marinaccio et al. [6] where a higher risk of injury on hot days was found among males and young (age 15–34) workers.

All the interviewees considered the average observable thermal damage, that is, they considered that the symptoms of injuries or illnesses due to heat exposure are on average recognizable. Meanwhile, the categories most at risk have little awareness of how preventive measures in the workplace can reduce the severity. The five main obstacles perceived by respondents to preventing heat-related injuries at work were lack of commitment by employers to protect health and safety, lack of training of company health and safety managers, lack of training of workers, lack of compliance with regulations, and lack of awareness among company health and safety managers on the risks deriving from heat stress.

As for the perception of risk knowledge, according to the entire sample, the scientific community has a fairly poor knowledge of heat risk, as do workers exposed to heat.

Consistently with the result of the perception of risk knowledge, the degree of knowledge of the heat risk resulting from this survey is low. Only one in four of the respondents received training on the prevention of heat-related injuries at work and an even lower proportion, 17.1%, received warnings or alarms.

The whole sample believed that heat is an important contributor to loss of productivity and this result is common in other surveys on the heat risk in the workplace. For example, Singh et al. [46], in a telephone survey carried out in Australia in the summer of 2010, focused on occupational heat risk, and showed that five dominant themes emerged on the effects of heat on the health and productivity of workers, one of them being the reduction in productivity due to heat.

To the best of our knowledge, this is the first study conducted at the national level in Italy to explore workers’ perception on the impact of heat stress on health, as well as to assess preventive practices and identify potential barriers to heat-related illnesses and injuries prevention in the workplace. While the COVID-19 pandemic hampered the conduction of case studies in the field in 2020, we were able to carry out a pilot study in preparation for the larger-scale surveys planned for the two subsequent summer seasons within the WORKLIMATE project.

Heat stress is an issue particularly for outdoor workers, and the latter represented the minority of participants in the 2020 survey. Unfortunately, the questionnaire submission during the COVID-19 pandemic, when many restrictions were in place in Italy also limiting outdoor activities, led to a prevalence of workers engaged in indoor activities among the respondents to the questionnaire. In the recruitment process, in the next survey iterations, it is crucial to increase the channels through which the questionnaire is distributed, to minimize selection bias and ensure outdoor workers who are most exposed are included. Nonetheless, information on awareness and perception of the problem of (mainly) indoor workers, allowed us to obtain useful information. The perception of indoors workers on heat stress is a seldom explored topic that needed to be evaluated.

Secondly, although the questionnaire had been built after taking into account functionally equivalent international and national questionnaires [19,22,23,24,25,26,28,29,30,31] and a pre-testing had been conducted on a random workers’ sample for optimization prior to the web-based survey launch, the pilot study allowed us to identify several questions that were too complicated and needed to be simplified and some others that were ambiguous or unnecessary and that needed to be discarded.

## 5. Conclusions

The survey highlighted that the sample of workers interviewed perceived a risk during a heat wave and that on average the heat risk does not depend on their wishes but can be potentially lethal. Unfortunately, however, some categories of workers, especially the youngest, still have a low perception of risk and this suggests the need to adopt policies to increase the risk perception related to heat. In addition, there is little awareness of how preventive measures in the workplace can reduce the severity of the heat risk and therefore the number of heat-related injuries were attributed by the majority of workers to the lack of training or in any case inadequate training; less than one in five workers received heat alarms. Although this survey represents only a sample of workers, with obvious limitations, especially regarding the low representation of outdoor workers, also because the COVID-19 restrictions during the pandemic period, highlights that Italian workers are not well prepared for the likelihood of increasing incidence of heat stress due to climate change. There is therefore a need to improve the heat risk prevention strategies in the occupational field by increasing training at multiple levels and developing appropriate heat health warning systems addressed to occupational sectors.

## Figures and Tables

**Figure 1 ijerph-19-08196-f001:**
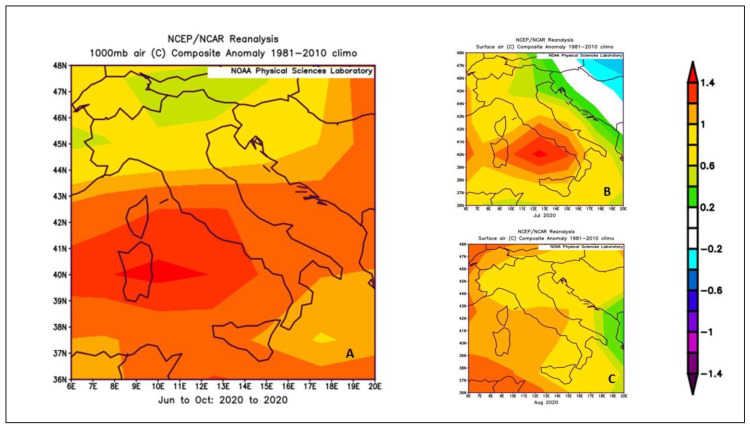
Air temperatures anomalies in Italy during the period June–October 2020 (**A**), July (**B**), and August 2020 (**C**) compared to the reference period 1981–2010. Data obtained from https://psl.noaa.gov/cgi-bin/data/composites/printpage.pl, accessed on 27 January 2022.

**Figure 2 ijerph-19-08196-f002:**
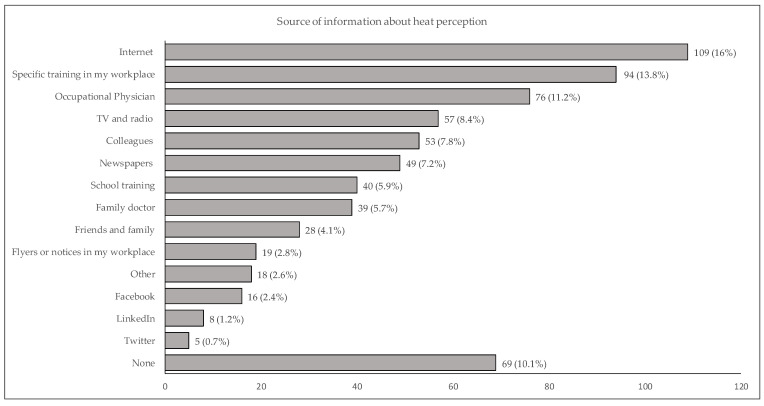
Frequencies and percentages of answers to the question 77—What are your main sources of information on the prevention of heat-related diseases in the workplace? (Multiple choice).

**Figure 3 ijerph-19-08196-f003:**
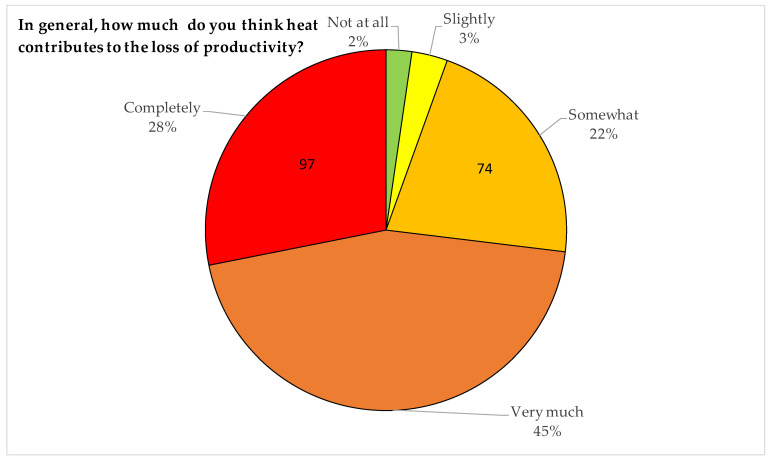
Frequencies and percentages of answers to the question 80—In general, how much do you think heat contributes to the loss of productivity?

**Figure 4 ijerph-19-08196-f004:**
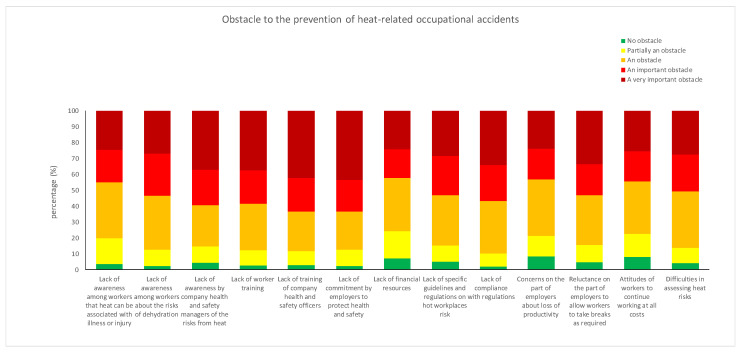
Percentages of answers to question 81—To what extent do you think that each of the following conditions can hinder prevention of heat-related occupational injuries? (A 5-point Likert scale from 1 = no obstacle at all to 5 = a very important obstacle).

**Table 1 ijerph-19-08196-t001:** Sample description.

		N	%
**Participants**		345	
**Gender**	Male	199	57.7
	Female	146	42.3
**Nationality**	Italian	331	95.9
	EU	11	3.2
	Non-EU	3	0.9
**Geographical area of working**	North	94	27.2
	Centre-South	251	72.8
**Marital status**	Married-Accompanied	201	58.3
	Other	144	41.7
**Age group**	0–34	62	18
	35–44	101	29.3
	45–54	113	32.8
	55+	69	20
**School degree qualification**	Primary school certificate	3	0.9
	Junior high school certificate	26	7.5
	High school diploma	105	30.4
	Bachelor’s degree	29	8.4
	Master’s degree/specialist degree	89	25.8
	Postgraduate training	93	27.0
**Workplace environment**	Mainly indoors in air-conditioning environment	224	64.9
	Mainly indoors in non-air-conditioned environment	73	21.2
	Mainly Outdoors	48	13.9
**Economic activity sector**	Agriculture, forestry, and fishing	5	1.4
	Extraction of minerals from quarries and mines	1	0.3
	Manufacturing	28	8.1
	Electricity, gas, steam, and air conditioning supply	3	0.9
	Water supply; sewerage, waste management, and remediation activities	3	0.9
	Construction-Building	54	15.7
	Trade	17	4.9
	Transport and storage	9	2.6
	Accommodation and food service activities	2	0.6
	Information and communication services	16	4.6
	Financial and insurance activities	13	3.8
	Real estate activities	1	0.3
	Professional, scientific, and technical activities	87	25.2
	Rental, travel agencies, business support services	1	0.3
	Public administration and defense	41	11.9
	Education	27	7.8
	Health and social work	28	8.1
	Artistic, sporting, entertainment, and recreational activities	9	2.6
**Number of employees in the company**	From 1 to 9 employees	79	22.9
	From 10 to 49 employees	63	18.3
	From 50 to 249 employees	89	25.8
	250 and more employees	114	33
**Intensity of physical activity in the workplace (on average)**	Very light-light	232	67.2
	Intense-very intense	113	32.8
**Heat sources**	Yes/sometimes	62	18
	No	283	82
**Use of chemicals**	Yes/sometimes	86	24.9
	No	259	75.1
**Wearing protective clothing**	Yes/sometimes	175	50.7
	No	170	49.3
**Use of COVID-19 face masks**	0 h	71	20.6
	From 1 to 5 h	160	46.4
	6 h and more	114	33
**Dealing with Occupational Safety and Health (OSH)**	Yes	206	59.7
	No	139	40.3
**Chronic diseases**	Yes	230	66.7
	No	115	33.3
**Injuries or accidents occurred during work experience due to hot/high humidity conditions**	Don’t know	32	9.3
	Never	90	26.1
	Rarely	100	29.0
	Few times	97	28.1
	Often	26	7.5
**Training on the prevention of heat-related injuries carried out in the workplaces**	Yes	53	15.4
	In some companies	35	10.1
	No	221	64.1
	Don’t know	36	10.4
**Warnings or alerts about the possibility of a heat wave received from employer**	No	286	82.9
	Yes, with messages	21	6.1
	Yes, verbally	24	7.0
	Yes, by notices placed at information points	4	1.2
	Yes, by company-specific training	10	2.9

**Table 2 ijerph-19-08196-t002:** Principal Component Analysis of section “Risk perception”. Extraction method: Principal Component Analysis. Rotation method: Varimax with Kaiser normalization.

N-Item	Component			
	1 “Personal Exposure and Fear of Risk”	2 “Collective Exposure and Risk Quality”	3 “Impact on Health and Prevention”	4 “Knowledge Risk Perception”
38—In summer, during my work, I feel exposed to heat (Personal exposure)	0.805			
41—I am afraid that heat waves will cause me to have an accident at work (Fear of risk)	0.781			
39—During a heat wave I feel very much at risk (Personal exposure)	0.780			
42—I am afraid that I will get sick because of heat waves (Fear of risk)	0.732			
29—Heat causes an immediate fatal effect for exposed persons (Immediacy effect)		0.754		
40—During a heat wave there are many workers at risk in Italy (Collective exposure)		0.709		
33—Heat is a potentially lethal risk (Chronic/Catastrophic)		0.693		
28—Workers are involuntarily exposed to heat (Voluntary risk)		0.538		
37—Heat risk damage is observable (Observability)			0.794	
36—Preventive measures in the workplace can reduce the severity of the heat risk (Controlling severity)			0.754	
31—The scientific world has a complete understanding of the heat risk (Knowledge of the risk)				0.819
30—Workers exposed to heat have precise knowledge of the risk (Knowledge of the risk)				0.731

**Table 3 ijerph-19-08196-t003:** Principal Component Analysis of section “Risk knowledge”. Extraction method: Principal Component Analysis.

N-Item	Component
	1 “Risk Knowledge”
48—People with heart disease are at risk of worsening their health during a heat wave	0.793
44—Heat can be the cause of accidents for outdoor workers	0.775
49. Heat-related illnesses can lead to death	0.772
45—Heat can cause injuries for those working in a non-air-conditioned indoor environment	0.747
50—Dehydration in hot weather predisposes to the development of serious kidney disease	0.692
54—Heat waves can be a risk factor for depression and anxiety	0.631

**Table 4 ijerph-19-08196-t004:** Means and standard deviations of the items in the section “Risk perception” on a 5-point Likert scale from 1 = “strongly disagree” to 5 = ”strongly agree”.

Risk Perception (Items)	Mean	SD
26—I feel that my health is threatened by climate change	3.22	1.01
27—I think that heat waves endanger my health	3.26	0.96
28—Workers are involuntarily exposed to heat	3.33	1.03
29—Heat causes an immediate fatal effect for those exposed	2.27	1.04
30—Workers exposed to heat have precise knowledge of the risk	2.20	0.84
31—The scientific world has a complete understanding of the heat risk	2.74	0.94
32—The heat risk is a new risk for Italian companies	2.98	1.07
33—Heat is a potentially lethal risk	3.32	0.99
34—Heat is a risk that workers have learned to live with	2.57	0.85
35—Heat poses a very low threat to future generations	1.77	0.95
36—Preventive measures in the workplace can reduce the severity of the heat risk	3.74	0.94
37—Heat risk damage is observable	3.36	0.93
38—In summer, during my work, I feel exposed to heat	2.96	1.10
39—During a heat wave I feel very much at risk	2.91	1.01
40—During a heat wave there are many workers at risk in Italy	3.66	0.85
41—I am afraid that heat waves will cause me to have an accident at work	2.65	1.15
42—I am afraid that I will get sick because of heat waves	2.43	1.05
43—During a heat wave I am afraid that the risk of transmission of the virus responsible for COVID-19 will increase	1.97	0.97

**Table 5 ijerph-19-08196-t005:** Personal exposure and fear of risk for three macro-groups (a demographic and professional characteristics, b characteristics of the work, c Factors aggravating heat stress) for the items 38, 39, 41, 42, 40 36, 31, 30. SD, Standard Deviation.

Demographic and Professional Characteristics Age Groups (Years)	N	%	Personal Exposure and Fear of Risk (N-Item)								Collective Exposure and Risk Quality (N-Item)		Impact on Health and Prevention (N-Item)		Knowledge of Risk Perception (N-Item)			
			38		39		41		42		40		36		31		30	
			Mean (SD)	F	Mean (SD)	F	Mean (SD)	F	Mean (SD)	F	Mean (SD)	F	Mean (SD)	F	Mean (SD)	F	Mean (SD)	F
≤40	103	29.9													2.57 (0.99)	4.64		
41–54	173	50.1													2.74 (0.86)			
≥55	69	20													3.01 (1.02)			
**School Degree**																		
Primary-high school diploma	134		3.29 (1.19)	19.65	3.15 (1.04)	13.01			2.26 (0.92)				3.52 (1.05)				2.38 (0.92)	
Bachelor’s degree-postgraduate training	211		2.74 (1.00)		2.75 (0.96)				2.48 (1.14)				3.88 (0.84)	11.11			2.08 (0.77)	9.82
**Job Years**																		
<5	84	24.3					2.49 (1.15)											
6–10	57	16.5					2.42 (1.08)											
11–20	104	30.1					2.56 (1.11)											
>21	100	29					3.00 (1.11)	4.75										
**Dealing with Occupational Safety and Health (OSH)**																		
Yes	206	59.7													2.86 (0.95)			
No	139	40.3													2.58 (0.91)	7.66		
**Characteristics of the Work** **Workplace Environment**	**N**	**%**	**Personal exposure and fear of risk (N-item)**								**Collective exposure and risk quality (N-item)**		**Impact on Health and Prevention** **(N-Item)**		**Knowledge of Risk Perception (N-Item)**			
			**38**		**39**		**41**		**42**		**40**		**36**		**31**		**30**	
			**Mean (SD)**	**F**	**Mean (SD)**	**F**	**Mean (SD)**	**F**	**Mean (SD)**	**F**	**Mean (SD)**	**F**	**Mean (SD)**	**F**	**Mean (SD)**	**F**	**Mean (SD)**	**F**
Mainly indoors in air-conditioning environment	224	64.9	2.58 (0.94)		2.77 (0.93)		2.45 (1.06)		3.10 (1.22)	10.77			3.83 (0.86)	6.31			2.08 (0.73)	10.08
Mainly indoors in non-air-conditioned environment	73	21.2	3.51 (1.06)		2.93 (1.06)		2.77 (1.22)						3.86 (0.89)	6.32			2.19 (0.84)	
Mainly Outdoors	48	13.9	3.85 (1.05)	47.74	3.50 (1.11)	10.87	3.38 (1.16)	14.23					3.15 (1.17)				2.75 (1.08)	
**Kind of Physical Activity in the Workplace (on Average)**																		
Very light-light	232	67.2	2.69 (0.99)				2.42 (1.05)						3.91 (0.81)	20.62			2.10 (0.78)	7.85
Intense-very intense	113	32.8	3.50 (1.13)	46.78			3.11 (1.22)	28.92					3.39 (1.09)				2.39 (0.94)	
**Training Heat-Related Injuries**																		
Yes/In some companies	88														3.08 (0.97)		2.42 (0.94)	
No/Don’t know	257														2.63 (0.91)	15.52	2.12 (0.79)	15.52
**Warnings Heat Wave Received**																		
No	286														2.67 (0.93)	10.48	2.14 (0.81)	7.13
Yes	59														3.10 (0.90)		2.49 (0.95)	
**Factors Aggravating Heat Stress** **Heat Sources**	**N**	**%**	**Personal Exposure and Fear of Risk (N-Item)**								**Collective Exposure and Risk Quality (N-Item)**		**Impact on health and prevention (N-item)**		**Knowledge of Risk Perception (N-Item)**			
			**38**		**39**		**41**		**42**		**40**		**36**		**31**		**30**	
			**Mean (SD)**	**F**	**Mean (SD)**	**F**	**Mean (SD)**	**F**	**Mean (SD)**	**F**	**Mean (SD)**	**F**	**Mean (SD)**	**F**	**Mean (SD)**	**F**	**Mean (SD)**	**F**
Yes/sometimes	62	18	3.63 (1.16)	30.37			3.24 (1.21)	21.38					3.39 (1.19)				2.48 (1.04)	
No	283	82	2.8 (1.04)				2.52 (1.10)						3.82 (0.86)	7.33			2.13 (0.78)	6.27
**Use of Chemicals**																		
Yes/sometimes	86	24.9	3.53 (1.19)	28.94			3.17 (1.16)	25.78										
No	259	75.1	2.76 (1.01)				2.47 (1.10)											
**Wearing Protective Clothing**																		
Yes/sometimes	175	50.7	3.30 (1.13)	38.87			3.01 (1.14)	39.64					3.57 (1.04)					
No	170	49.3	2.60 (0.96)				2.27 (1.04)						3.92 (0.79)	12.08				
**Use of COVID-19 masks**																		
0 h	71	20.6	2.72 (1.06)	5.15														
From 1 to 5 h	160	46.4	2.88 (1.10)															
6 h and more	114	33	3.21 (1.11)															
**Chronic Diseases**																		
Yes	230	66.7			3.15 (1.07)	10.04					3.83 (0.76)	8.09						
No	115	33.3			2.79 (0.96)						3.57 (0.88)							

## Data Availability

Not applicable.

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
