# Peer review of "Workers’ Perception Heat Stress: Results from a Pilot Study Conducted in Italy during the COVID-19 Pandemic in 2020"

_ijerph, 2022, doi:10.3390/ijerph19138196_

Round 1

Reviewer 1 Report

The authors present a study about workers' risk perception of heat stress using a cross-sectional online questionnaire survey conducted in Italy in the hottest months of 2020 (June-October) through different multimedia channels.  The authors have the intention to identify workers’ needs and gaps in knowledge and suggest the adaptation of measures that best comply with the needs of both workers and employers.  It is a very interesting paper and in general well written.

However, it is often considered studies use exposure limits intended to minimize the risk of heat-related illness using WBGT ºC (wet bulb globe temperature), which takes into account environmental factors, such as air temperature, humidity and air movement, which contribute the to the perception of hotness by people. In some workplace situations, solar load (heat from radiant sources) is also considered in determining the WBGT. So, the question is how the workers' risk perception of heat stress studied, can be related to real heat conditions in the same period. Have the authors measured the heat stress parameters?

Improvements in some parts of the documents are suggested, mainly in tables/figures identified below:

-          The table 1 must be improved. It is difficult to read, so the authors can present a new organization.

-          The Figure 3 can be reduced.

-          The table 3 must be improved also, reorganize for better reading.

Author Response

  1. The authors present a study about workers' risk perception of heat stress using a cross-sectional online questionnaire survey conducted in Italy in the hottest months of 2020 (June-October) through different multimedia channels. The authors have the intention to identify workers’ needs and gaps in knowledge and suggest the adaptation of measures that best comply with the needs of both workers and employers. It is a very interesting paper and in general well written.

However, it is often considered studies use exposure limits intended to minimize the risk of heat-related illness using WBGT ºC (wet bulb globe temperature), which takes into account environmental factors, such as air temperature, humidity and air movement, which contribute the to the perception of hotness by people. In some workplace situations, solar load (heat from radiant sources) is also considered in determining the WBGT. So, the question is how the workers' risk perception of heat stress studied, can be related to real heat conditions in the same period. Have the authors measured the heat stress parameters?

Answer 1: First of all, we thank the reviewer for the compliments, for the suggestions and criticisms provided, because they are a great opportunity for us to improve the paper and make it more understandable to readers. As regards the heat risk perception, in this study we did not combine the administration of the questionnaire with a direct measurement of WBGT. This is because the questionnaire was largely administered online and its objective is not to evaluate the heat risk perception linked to the microclimatic conditions during the questionnaire compilation but rather to quantify the awareness of the heat risk in the summer period by the worker. The heat risk perception and the assessment of the environmental thermal stress conditions were carried out within the worklimate project but in specific case studies carried out in specific days on a small sample of workers in some companies selected in the agricultural and construction sector. In these case studies, a specific questionnaire was administered to assess the thermal stress conditions, physiological measurements were carried out on the worker (body temperature, oxygen saturation, heart rate) and a continuous microclimatic monitoring was carried out with calculation of hourly WBGT. Furthermore, the survey analyzed in this work was mainly administered online to workers who perform tasks in an indoor environment, therefore the hypothesis of using a WBGT data obtained from a meteorological model (reanalysis) was not even considered.

  1. The table 1 must be improved. It is difficult to read, so the authors can present a new organization

Answer 2: We have reorganized the table in order to improve understanding.

N

%

Participants

345

Gender

Male

199

57.7

Female

146

42.3

Nationality

Italian

331

95.9

EU

11

3.2

Non-EU

3

0.9

Geographical area of working

North

94

27.2

Centre-South

251

72.8

Marital status

Married-Accompanied

201

58.3

Other

144

41.7

Age group

0-34

62

18

35-44

101

29.3

45-54

113

32.8

55+

69

20

School degree qualification

Primary school certificate

3

0.9

Junior high school certificate

26

7.5

High school diploma

105

30.4

Bachelor's degree

29

8.4

Master's degree/specialist degree

89

25.8

Postgraduate training

93

27.0

Workplice environment

Mainly indoors in air-conditioning environment

224

64.9

Mainly indoors in non-air-conditioned environment

73

21.2

Mainly Outdoors

48

13.9

Economic activity sector

Agriculture, forestry and fishing

5

1.4

Extraction of minerals from quarries and mines

1

0.3

Manufacturing

28

8.1

Electricity, gas, steam and air conditioning supply

3

0.9

Water supply; sewerage, waste management and remediation activities

3

0.9

Construction – Building

54

15.7

Trade

17

4.9

Transport and storage

9

2.6

Accommodation and food service activities

2

0.6

Information and communication services

16

4.6

Financial and insurance activities

13

3.8

Real estate activities

1

0.3

Professional, scientific and technical activities

87

25.2

Rental, travel agencies, business support services

1

0.3

Public administration and defence

41

11.9

Education

27

7.8

Health and social work

28

8.1

Artistic, sporting, entertainment and recreational activities

9

2.6

Number of employees in the company

From 1 to 9 employees

79

22.9

From 10 to 49 employees

63

18.3

From 50 to 249 employees

89

25.8

250 and more employees

114

33

Intensity of physical activity in the workplace (on average)

Very light-light

232

67.2

Intense-very intense

113

32.8

Heat sources

Yes/sometimes

62

18

No

283

82

Use of chemicals

Yes/sometimes

86

24.9

No

259

75.1

Wearing protective clothing

Yes/sometimes

175

50.7

No

170

49.3

Use of COVID-19 face masks

0 hours

71

20.6

From 1 to 5 hours

160

46.4

6 hours and more

114

33

Dealing with Occupational Safety and Health (OSH)

Yes

206

59.7

No

139

40.3

Chronic diseases

Yes

No

230

66.7

115

33.3

Injuries or accidents occurred during work experience due to hot/high humidity conditions

Don't know

32

9.3

Never

90

26.1

Rarely

100

29.0

Few times

97

28.1

Often

26

7.5

Training on the prevention of heat-related injuries carried out in the workplaces

Yes

53

15.4

In some companies

35

10.1

No

221

64.1

Don't know

36

10.4

Warnings or alerts about the possibility of a heat wave received from employer

No

286

82.9

Yes, with messages

21

6.1

Yes, verbally

24

7.0

Yes, by notices placed at information points

4

1.2

Yes, by company-specific training

10

2.9

  1. The Figure 3 can be reduced.

Answer 3: We have reduced Figure 3.

  1. The table 3 must be improved also, reorganize for better reading.

Answer 4: Concerning table 3, it shows the results of the Principal Component Analysis of the “Risk knowledge” section. Since it is a single factor analysis, compared to table 2, it shows only one column. Within the column, the items are sorted by saturation.

Reviewer 2 Report

 The authors present the findings of a nationwide questionnaire-based survey conducted in Italy to assess employees' perceptions and understanding of the negative effects of occupational heat stress, particularly during the covid epidemic when workers wore protective masks.

The authors attempted to identify potential educational and training barriers, as well as issues that must be addressed, in order to successfully prevent heat-related illnesses in the workplace. The study's ultimate goal was to identify workers' needs and knowledge gaps in order to provide adaptation solutions that best meet the needs of both workers and employers.

 This interesting work needs major revision, these are my comments and suggestions:

Lines 64-74 The authors have six references at the end of this 10-line paragraph. It would be preferable to place some of the references in the appropriate line to specify what is supported by which reference.

 Line 70.Please cite using the appropriate reference style.

 Line 185. Please describe the "groups."

 Line 286. Table 5 contains a massive amount of parameters and statistical analysis.  I urge the authors to help the readers by dividing the table into sections; for example, the analysis for the various demographic "groups" can be presented in a single Table.  In any case, present the possible correlations and significance of parameters/differences; the format should be determined by the authors based on what they want to emphasize and help the reader see in their results. Table 5 is currently not assisting readers in this manner. This suggestion is crucial for me, I need to see what the authors can conclude on the basis of this analysis of the data presented in Table 5. This should be one of the major issues to address in the revision of this manuscript.

Line 412. I'm curious if the authors can look into the effect of time of day (among other parameters temperature peaks at certain time of the day). Workers' perceptions of the risk of exposure to elevated thermal conditions may vary depending on the hours worked, and the risk of occupational accidents may also vary depending on the season and time of day, see for example "TILIGADAS, I., et al. A Review of Health and Safety Issues in the Greek Mariculture Industry." Journal of Scientific Research and Reports, 2014, 3.9: 1153-1161, doi:10.9734/JSRR/2014/8886."

-Fatima, S. H., Rothmore, P., Giles, L. C., Varghese, B. M., & Bi, P. (2021). Extreme heat and occupational injuries in different climate zones: A systematic review and meta-analysis of epidemiological evidence. Environment international, 148, 106384.
-Fatima, S. H., Rothmore, P., Giles, L. C., & Bi, P. (2022). Outdoor ambient temperatures and occupational injuries and illnesses: Are there risk differences in various regions within a city?. Science of the total environment, 826, 153945.
-Nizam, C. M., Ismail, A. R., Sukadrin, E. H., Mokhtar, N. K., Abdullah, A., Jusoh, N., & Husshin, N. (2022). A Short Review on Heat Stress and Heat Strain in Construction Industry: The Effect on Worker Performance, Associated Health Effect, It’s Measurement and Control Mechanism. Human-Centered Technology for a Better Tomorrow, 559-566.

 Line 426, 448, and 466  Why bring up the pilot study at this point in the discussion? If you must do this, please do so much earlier, perhaps in the Methodology section.

Author Response

Referee 2:

  1. The authors present the findings of a nationwide questionnaire-based survey conducted in Italy to assess employees' perceptions and understanding of the negative effects of occupational heat stress, particularly during the covid epidemic when workers wore protective masks. The authors attempted to identify potential educational and training barriers, as well as issues that must be addressed, in order to successfully prevent heat-related illnesses in the workplace. The study's ultimate goal was to identify workers' needs and knowledge gaps in order to provide adaptation solutions that best meet the needs of both workers and employers.

Answer 1: we thank the reviewer for the appreciation for the study we have carried out. The comments were very helpful in further improving the paper and making it easier.

Lines 64-74. The authors have six references at the end of this 10-line paragraph. It would be preferable to place some of the references in the appropriate line to specify what is supported by which reference.

Answer 2: Thank you for your suggestion. We have moved the references to the exact point they refer to.

Line 70. Please cite using the appropriate reference style.

Answer 3: Sorry, the quote was a typo and we removed it. This quote can be found in the discussion section instead (45).

Line 185. Please describe the "groups."

Answer 4: We thank the reviewer for giving us the opportunity to better explain what we carried out. We have now added in the text some examples of analyzed groups and above all the creation of 3 macro-groups: “The chosen groups (for example age, school degree qualification, workplace environment, use of wearing protecting clothing, use of covid 19 mask, chronic diseases, etc.) were further grouped into 3 macrogroups (a. Demographic and professional characteristics, b. Characteristics of the work, c. Factors aggravating heat stress) in order to evaluate the fundamental aspects in the assessment of risk per-ception.”

Line 286. Table 5 contains a massive amount of parameters and statistical analysis.  I urge the authors to help the readers by dividing the table into sections; for example, the analysis for the various demographic "groups" can be presented in a single Table.  In any case, present the possible correlations and significance of parameters/differences; the format should be determined by the authors based on what they want to emphasize and help the reader see in their results. Table 5 is currently not assisting readers in this manner. This suggestion is crucial for me, I need to see what the authors can conclude on the basis of this analysis of the data presented in Table 5. This should be one of the major issues to address in the revision of this manuscript.

Answer 5: we thank the reviewer because we actually realized that table 5 was not well organized. We therefore proceeded to break down table 5 into 3 macrogroups (a. Demographic and professional characteristics, b. Characteristics of the work, c. Factors aggravating heat stress) to highlight the three main factors. This certainly helps the reader to understand the subsequent choice of groups. we have also added this aggregation in macro-groups also in the materials and methods, in order to explain to the reader, the type of analysis that has been carried out.

Line 412. I'm curious if the authors can look into the effect of time of day (among other parameters temperature peaks at certain time of the day). Workers' perceptions of the risk of exposure to elevated thermal conditions may vary depending on the hours worked, and the risk of occupational accidents may also vary depending on the season and time of day, see for example "TILIGADAS, I., et al. A Review of Health and Safety Issues in the Greek Mariculture Industry." Journal of Scientific Research and Reports, 2014, 3.9: 1153-1161, doi:10.9734/JSRR/2014/8886."

-Fatima, S. H., Rothmore, P., Giles, L. C., Varghese, B. M., & Bi, P. (2021). Extreme heat and occupational injuries in different climate zones: A systematic review and meta-analysis of epidemiological evidence. Environment international, 148, 106384.

-Fatima, S. H., Rothmore, P., Giles, L. C., & Bi, P. (2022). Outdoor ambient temperatures and occupational injuries and illnesses: Are there risk differences in various regions within a city?. Science of the total environment, 826, 153945.

-Nizam, C. M., Ismail, A. R., Sukadrin, E. H., Mokhtar, N. K., Abdullah, A., Jusoh, N., & Husshin, N. (2022). A Short Review on Heat Stress and Heat Strain in Construction Industry: The Effect on Worker Performance, Associated Health Effect, It’s Measurement and Control Mechanism. Human-Centered Technology for a Better Tomorrow, 559-566.

Answer 6: Your consideration is undoubtedly interesting and we too have raised the problem to us. The survey presented in this paper constitutes only one of the activities of the WORKLIMATE project and among the many other activities there are also case studies carried out within sample companies in the agricultural and construction sector. In these case studies, microclimatic, physiological and behavioral monitoring were carried out on sample subjects on specific hot risk days. During these days, questionnaires on thermal perception were also administered at three times of the day (in the morning, central hours of the day, afternoon) in order to assess the impact of microclimatic conditions also on thermal perception. The data collected is being processed and will soon be published in a new publication in which we will certainly also mention the works you have suggested.

Line 426, 448, and 466  Why bring up the pilot study at this point in the discussion? If you must do this, please do so much earlier, perhaps in the Methodology section.

Answer 7: Regarding the paragraph starting at line 426 and 448 we agree with you that this information can be deleted as it is already in materials and methods and does not add further information to the discussion. The final consideration (line 466) I believe that it can be left in this section because it highlights the importance of repeated administrations for the calibration of increasingly performing questionnaires. In any case, we are talking about a pilot study from the beginning of the paper, this is because this survey will be re-proposed again next year to a greater number of working sectors, in order to improve the sample.
